# Characterization of the spatiotemporal representations of visual, semantic, and memorability features in the human brain

Yue Wang[ID][1]*, Peter Brunner[2], Jon T. Willie[2], Runnan Cao[1†]*, Shuo Wang[ID][1,2†]*

1 Department of Radiology, Washington University in St. Louis, St. Louis, Missouri, United States of America, 2 Department of Neurosurgery, Washington University in St. Louis, St. Louis, Missouri, United States of America

† These authors are co-senior authorship on this work.
* yue.w@wustl.edu (YW); r.cao@wustl.edu (RC); shuowang@wustl.edu (SW)

## Abstract

Object recognition requires integrated processing that extends beyond the visual cortex, incorporating semantic and memory-related processes. However, it remains unclear how different attributes, such as visual, semantic, and memorability features, are encoded and interact during perception. Here, we recorded intracranial electroencephalography from 5,143 channels while participants viewed natural object images. We systematically characterized the spatiotemporal patterns of neural encoding for visual, semantic, and memorability attributes and showed that memorability was encoded in a distributed manner, which can be dissociated from visual and semantic coding. While the ventral temporal cortex (VTC) was engaged in encoding all three attributes, the representations were dissociable. Interestingly, memorability representations in the prefrontal cortex appeared to arise from integrated visual and semantic signals from the VTC; and memorability influenced early stages of visual and semantic processing. Our results were corroborated by high-resolution 7T fMRI, which revealed continuous encoding across the brain, and further validated using a separate dataset featuring within-category object variability. Lastly, single-neuron recordings confirmed semantic and memorability coding in the medial temporal lobe. Together, these findings provide a comprehensive view of how visual, semantic, and memorability attributes are dynamically encoded across the brain, highlighting the complex interplay between these attributes that collectively shape object recognition and memory formation.

## Introduction

A remarkable capacity of the human brain is its ability to encode and retain detailed visual memories after just a single exposure [1–3]. However, the neural mechanisms that support such vivid memory formation remain incompletely understood. The

**Data availability statement:** All data and code that support the findings of this study are publicly available on OSF (https://osf.io/muq48/). Source data for each figure are uploaded as S1 Data–S8 Data.

**Funding:** This research was supported by the NIH (https://www.nih.gov/; K99EY036650 [R.C.], R01MH129426 [S.W.], R01MH120194 [J.T.W.], R01EB026439 [P.B.], U24NS109103 [P.B.], U01NS108916 [P.B.], U01NS128612 [P.B.], R21NS128307 [P.B.], P41EB018783 [P.B.]), AFOSR (https://www.afrl.af.mil/AFOSR/; FA9550-21-1-0088 [S.W.]), NSF (https://www.nsf.gov/; BCS-1945230 [S.W.]), Brain & Behavior Research Foundation (https://bbrfoundation.org/; 33261 [R.C.]), and McDonnell Center for Systems Neuroscience (https://sites.wustl.edu/systemsneuroscience/; [R.C.]). The funders had no role in study design, data collection and analysis, decision to publish, or preparation of the manuscript.

**Competing interests:** The authors have declared that no competing interests exist.

**Abbreviations:** DNN, deep neural network; FG, fusiform gyrus; GLME, generalized linear mixed-effects; HGP, high-gamma power; iEEG, intracranial electroencephalography; LLM, large language model; MTL, medial temporal lobe; PCA, principal component analysis; PFC, prefrontal cortex; PLS, partial least squares; RDM, representational dissimilarity matrix; ROI, region of interest; RSA, representational similarity analysis; SNR, signal-to-noise ratio; VTC, ventral temporal cortex.

exceptional ability relies on the integration of visual and semantic processing, which interacts with the intrinsic memorability of object images. The concept of "memorability" arises from the observation that certain images are consistently more memorable than others, even within the same category [4,5]. This effect is robust across individuals and constitutes an isolable phenomenon that cannot be explained solely by attention, semantics, or low-level visual features [6–8]. Recently, understanding variations in the memorability of visual images has garnered increasing attention, as it is critically linked to visual memory and the development of reliable measures for predicting memory performance [8,9].

Neural correlates of memorability have been identified in multiple brain areas, including the ventral temporal cortex (VTC), superior temporal cortex, medial temporal lobe (MTL), and parietal cortex [10–14]. Notably, highly memorable images are more decodable and elicit stronger and more persistent neural responses than low-memorability images [12,14,15], suggesting that memorability facilitates perceptual processing and thus contributes to better memory performance. Importantly, neural substrates of memorability are dissociable from those of individual memory, indicating that the brain may represent memorability as a distinct attribute—separate from whether a specific image is remembered—within partially independent perceptual and mnemonic streams [10,12,16]. Such a representation could confer adaptive value: by preferentially enhancing processing of intrinsically memorable stimuli, the brain may optimize storage of information that is statistically more likely to be behaviorally relevant or socially shared. Our recent study has shown that single neurons in the human MTL not only demonstrate memory selectivity (i.e., differentiating novel versus familiar stimuli) but also correlate parametrically with the memorability scores of images [17], reinforcing the view that memorability is an intrinsic, image-dependent property with a dedicated neural signature. Together, these findings support the idea that memorability is not merely an epiphenomenon of perception or memory, but a neural feature with functional significance for visual memory systems.

Despite these findings, it remains unclear what makes some images more memorable than others [6] and how memorability interacts with perception and memory at the neural level. Earlier studies primarily linked the memory fate of stimuli to attentional states and task context, utilizing individual-centric methods [18–20]. However, recent advances in deep learning algorithms and the availability of large-scale online behavioral studies have facilitated stimulus-centric research, revealing that visual and semantic features embedded in object images can predict memorability scores [21–23]. Recent research has demonstrated that memorability is represented in both higher visual cortex and mnemonic regions, where it may interact with high-level visual and semantic factors [14,24]. However, it remains unknown whether memorability is represented as a distinct neural code or emerges from the interplay of visual and semantic processing. In particular, the mechanisms through which the neural processing of the three interrelated attributes—visual, semantic, and memorability—culminates in the formation of visual memory have yet to be elucidated. Specifically, the higher visual cortex subserves the neural representations of visual [25], semantic [26], and memorability [14] features. Are these representations independent of one

another, or are they interconnected within the spatiotemporal dynamics of the higher visual cortex and other associated brain regions? Elucidating these questions requires high spatiotemporal resolution signals to enable fine-grained analysis within specific brain areas, particularly the anterior VTC and MTL, which are prone to signal loss in neuroimaging studies [27,28].

To address these questions, we recorded intracranial electroencephalography (iEEG) activity across a broad range of human brain areas, including the VTC, MTL, and prefrontal cortex (PFC), while neurosurgical patients viewed 500 naturalistic object images spanning 50 diverse categories. To systematically investigate these questions, we employed representational similarity analysis (RSA) to compare the visual, semantic, and memorability representations with neural signals across spatiotemporal dimensions. Specifically, we first mapped the spatiotemporal distribution of neural representations for visual, semantic, and memorability features separately. Next, we examined how these representations relate to each other. Third, we characterized the dynamic flow of information across different brain areas. To validate and generalize our findings, we used an additional dataset comprising more complex and naturalistic object images. Furthermore, recognizing that neural signals at different scales may encode distinct types of information, we explored these questions at both the microscopic level of single-neuron activity and the macroscopic level of fMRI signals.

## Results

### Neural encoding of visual, semantic, and memorability attributes across the brain

Twenty neurosurgical patients (12 females; S1 Table) undergoing intracranial monitoring with implanted depth electrodes participated in this study. We recorded iEEG activity while participants viewed 500 ImageNet images [29], in which they were instructed to respond whenever an image was repeated in two consecutive trials (Fig 1A; see Methods for details). Participants demonstrated high task engagement, with a mean accuracy of 87.53% ± 10.82% (mean ± SD) across sessions.

We recorded iEEG from a total of 5,143 channels (Fig 1B), covering the VTC ($n = 924$), superior temporal gyrus (STG; $n = 551$), middle temporal gyrus (MTG; $n = 565$), MTL ($n = 668$), and PFC ($n = 356$). These brain areas were designated as regions of interest (ROIs). For analysis, we extracted high-gamma power (HGP; 70–170 Hz) from each iEEG channel, as HGP reflects the average neuronal firing of local neural populations [30,31]. We identified a total of 1,678 visually responsive channels (black dots in Fig 1B) by comparing neural responses around peaks (0.1 to 0.6 s relative to stimulus onset; Fig 1C) to baseline activity (−0.5 to 0 s relative to stimulus onset), including 462 channels in the VTC, 120 channels in the STG, 118 channels in the MTG, 285 channels in the MTL, and 112 channels in the PFC. Subsequent analyses were restricted to these visually responsive channels. Furthermore, we qualitatively observed that fusiform gyrus (FG) channels responded more strongly to human images (in the "gymnast" category) and animal images (S1A Fig), consistent with classical views of visual category selectivity in the VTC [32–34] (note that population responses in the VTC also reflected categorical structure; S1B Fig).

To investigate how the brain encodes visual, semantic, and memorability features, we constructed representational dissimilarity matrices (RDMs) across object images using features extracted from the output layers of three distinct deep neural networks (DNNs), each carrying visual, semantic, or memorability-related embeddings in their latent spaces (see Methods for details; see S2A, S2C and S2E Fig for analyses across layers). These DNNs, well established in the literature, included ResNet-101 [35] for visual object categorization, SGPT [36] for semantic search, and ResMem [37] for memorability score prediction (see S2B, S2D and S2F Fig for analyses using different models). Visualizing the extracted features in two-dimensional spaces revealed a highly meaningful arrangement of object images, organized by visual, semantic, or memorability features (Fig 1D). Consistent with previous findings [15,23,38], the representations of visual, semantic, and memorability features in the DNNs were inherently correlated with one another (Fig 1E), despite being derived from different models.

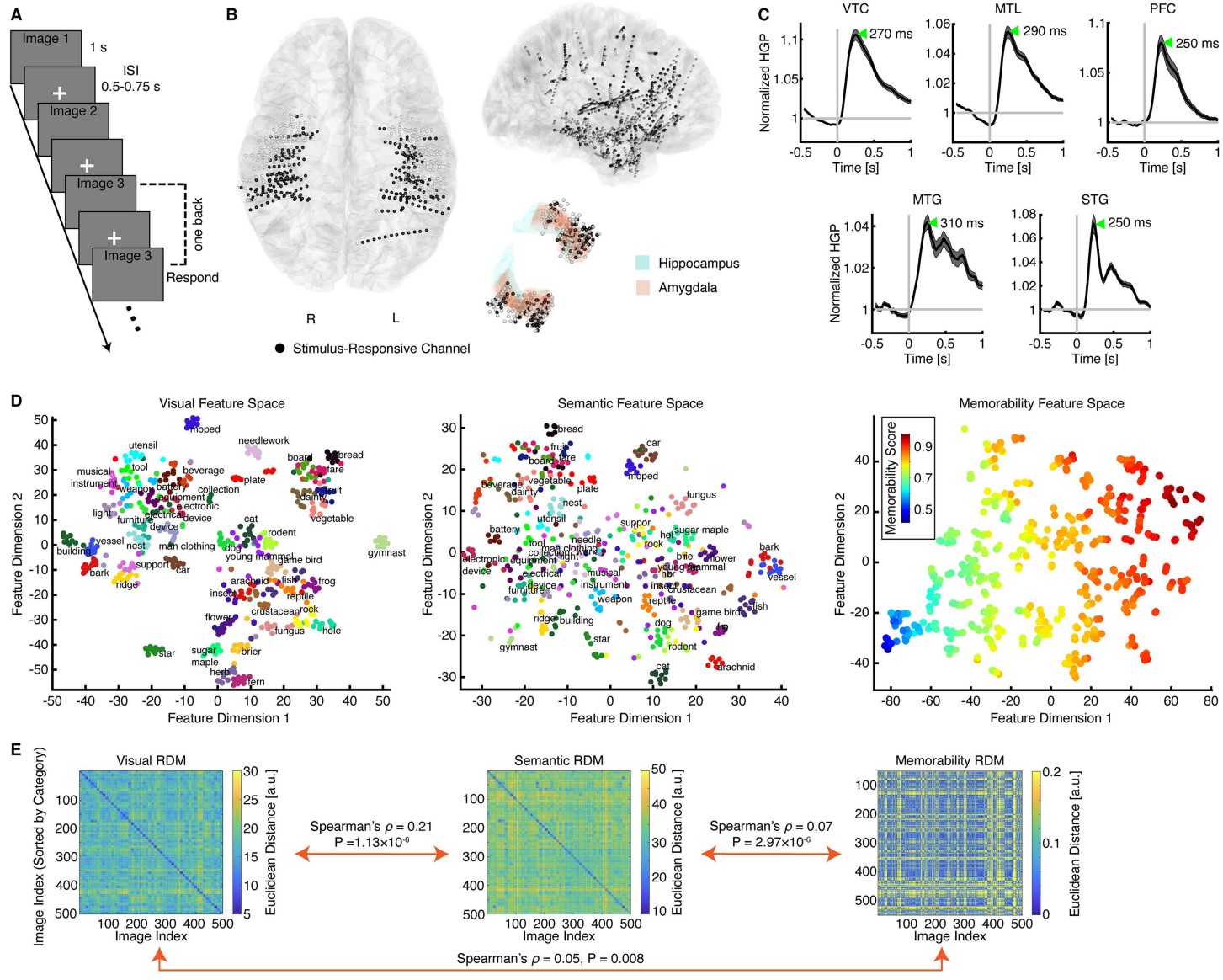

**Fig 1. Experimental procedures. (A)** Task. We presented 500 different object images to the participants in a randomized order during a one-back task. Participants were instructed to press a button whenever an identical image was repeated consecutively. Each image was displayed for 1 s, followed by a jittered inter-stimulus interval (ISI) ranging from 0.5 to 0.75 s. **(B)** Electrode localization. Electrodes from all participants were projected onto the MNI-305 template brain. Black: visually responsive channels. **(C)** Time course of representative channels in each region of interest (ROI). Error shades indicate ±SEM across trials. The green arrow indicates the peak latency. **(D)** Visualization of the feature spaces. Category labels are displayed at the median coordinates of each category in the visual and semantic spaces. Colors indicate categories (left and middle) or memorability scores (right). **(E)** Representational dissimilarity matrices (RDMs). RDMs were constructed using visual, semantic, and memorability features, illustrating the dissimilarity structures derived from the feature spaces. The source data underlying this figure are provided in S1 Data.

We first assessed whether individual channels encoded visual, semantic, and memorability features separately by correlating each channel's RDM with the respective feature RDMs. We observed a significant percentage of channels encoding visual features in the VTC ($n = 63$, 13.64%, binomial test against 5% chance level: $P = 3.74 \times 10^{-13}$; Fig 2A and 2B), semantic features in the VTC ($n = 51$, 11.04%, binomial $P = 6.24 \times 10^{-8}$) and MTL ($n = 31$, 10.88%, binomial

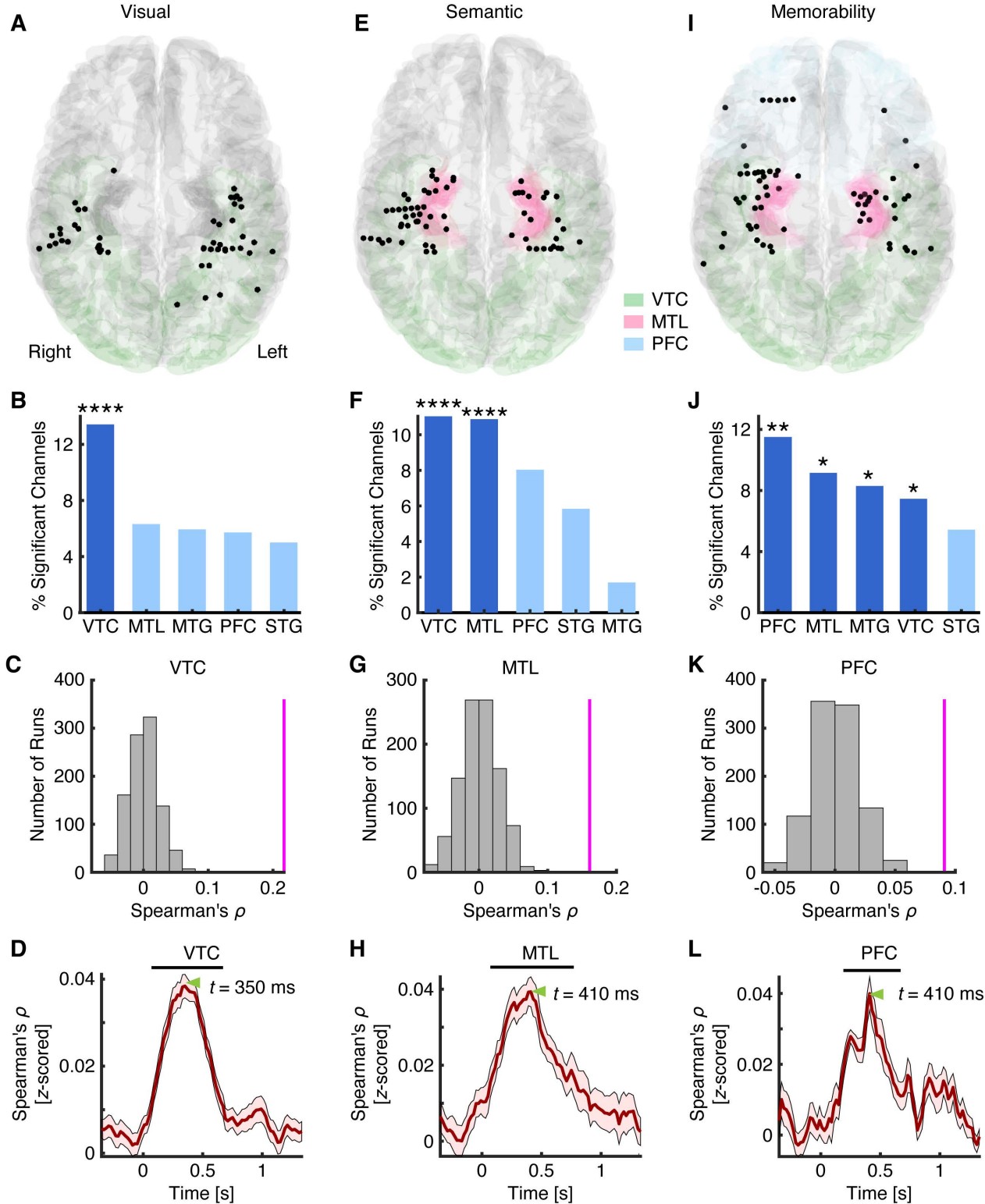

**Fig 2. Encoding of visual, semantic, and memorability features. (A–D)** Visual coding. **(E–H)** Semantic coding. **(I–L)** Memorability coding. **(A, E, I)** Spatial distribution of significant channels encoding **(A)** visual, **(E)** semantic, and **(I)** memorability features. Channels were projected onto the MNI brain space. Each dot represents a channel. Color coding indicates different brain areas. **(B, F, J)** Percentage of significant channels in each region of interest

(ROI). Group-level significance was determined using a binomial test. *: $P<0.05$, **: $P<0.01$, ***: $P<0.001$, and ****: $P<0.0001$. Dark colors represent an above-chance number of selected channels in the corresponding ROIs (binomial test: $P<0.05$), while light colors indicate chance-level selection. **(C, G, K)** Population-level neural encoding of visual, semantic, and memorability features in the most prominent ROI associated with each attribute. Magenta lines indicate the observed correlation coefficients, while gray bars represent the null distribution estimated via permutation (1,000 runs). **(D, H, L)** Temporal profiles of neural representations for the **(D)** visual, **(H)** semantic, and **(L)** memorability spaces. Black bars indicate time points with significant encoding, and green triangles mark the peaks of encoding. MTG: middle temporal gyrus. MTL: medial temporal lobe. PFC: prefrontal cortex. STG: superior temporal gyrus. VTC: ventral temporal cortex. The source data underlying this figure are provided in S2 Data.

$P=1.99\times10^{-5}$; Fig 2E and 2F), and memorability features in the PFC ($n=12$, 10.71%, binomial $P=0.004$), MTL ($n=22$, 7.72%, binomial $P=0.017$), MTG ($n=10$, 8.47%, binomial $P=0.035$), and VTC ($n=32$, 6.93%, binomial $P=0.035$; Fig 2I and 2J). Our results remained robust across different layers of the same model (S2A, S2C and S2E Fig), as well as across different models (S2B, S2D and S2F Fig). We also obtained consistent results using cosine distance rather than Euclidean distance (S2G–S2I Fig). Furthermore, to account for the fact that a subset of patients underwent two recording sessions (S1 Table), we repeated all analyses using only the first session from each participant and obtained consistent results (S2M–S2O Fig). Therefore, by examining the neural correlates of these three attributes collectively and comprehensively, we revealed their differential distributions in the human brain: visual information is represented primarily in the VTC, semantic information is represented mainly in the VTC and MTL, and memorability processing engages a broader distribution of higher-level brain areas beyond the VTC. Our findings also align with previous studies identifying neural correlates in the VTC for visual, semantic, and memorability attributes, albeit from independent investigations [14,39,40].

To further evaluate whether PFC memorability encoding persists after accounting for the nested structure of the intracranial dataset, we expanded our analysis to include all PFC channels using linear mixed-effects models. This approach uses the continuous individual-channel RSA correlation coefficient (see Methods) as a sensitive measure of memorability encoding and explicitly models participants and/or sessions as random effects. Consistent with the above findings, memorability encoding in PFC channels remained significant when modeling participants (correlation coefficient ~ 1 + (1 | participant); $P=0.0052$), sessions (correlation coefficient ~ 1 + (1 | participant:session); $P=0.0052$), and sessions nested within participants (correlation coefficient ~ 1 + (1 | participant) + (1 | participant:session); $P=0.047$) as random effects. Furthermore, both a bootstrapping test and a generalized linear mixed-effects (GLME) model indicated that individual differences in the percentage of memorability-coding PFC channels were not significant (both $P$s $>0.05$; see Methods for details). Therefore, these results demonstrate that PFC memorability encoding is robust even after appropriately accounting for the nested nature of the data.

Both nonhuman primate studies [41,42] and our recent iEEG studies [43,44] have demonstrated robust linear encoding of visual features in the VTC. However, it remains unclear whether the VTC and other brain areas encode semantic and memorability features using similar linear models, which may not be fully captured by the RDM-based analysis described above. To address this, we fitted a linear regression model (see Methods for details) to each channel separately using the three types of features: visual, semantic, and memorability. The results were qualitatively consistent with the RDM model, as we observed robust encoding of visual (S2J Fig; $n=157$, 33.98%, binomial $P=7.14\times10^{-14}$), semantic (S2K Fig; $n=129$, 27.92%, binomial $P=7.13\times10^{-14}$), and memorability (S2L Fig; $n=144$, 31.17%, binomial $P=7.13\times10^{-14}$) features in the VTC. The percentage of selective channels identified by the linear model was even significantly higher than those identified by the RDM model ($\chi^2$-test; visual: $P=3.85\times10^{-13}$; semantic: $P=9.23\times10^{-11}$; memorability: $P<1\times10^{-20}$). Furthermore, we observed a significant number of channels encoding visual ($n=51$, 17.89%, binomial $P=1.41\times10^{-13}$), semantic ($n=38$, 13.33%, $P=1.55\times10^{-8}$), and memorability ($n=31$, 10.88%, binomial $P=1.99\times10^{-5}$) features in the MTL; and we also observed a significant number of channels encoding visual ($n=12$, 10.71%, binomial $P=0.004$) and memorability ($n=14$, 12.5%, binomial $P=4.79\times10^{-4}$) features in the PFC.

PLOS Biology

Do these brain areas represent the structure of visual (Fig 1E, left), semantic (Fig 1E, middle), and memorability (Fig 1E, right) information at the population level? To address this question, we computed RDMs using all selective channels within the most prominent ROI associated with each attribute (visual, semantic, or memorability) and performed RSA with the corresponding feature RDMs. Indeed, we observed significant representation of visual features in the VTC (Fig 2C; Spearman's $\rho = 0.22$, permutation $P < 0.001$), semantic representation in the MTL (Fig 2G; Spearman's $\rho = 0.16$, permutation $P < 0.001$), and memorability representation in the PFC (Fig 2K; Spearman's $\rho = 0.09$, permutation $P < 0.001$) at the population level. We also derived similar results using all visually responsive channels (i.e., not limited to those selective for a given attribute; S1B Fig; all permutation $P$s $< 0.05$).

How are visual, semantic, and memorability features encoded *temporally* in their corresponding dominant brain areas? To address this, we conducted time-resolved RSA on the selected channels within each ROI (see Methods for details). Consistent with previous literature [45], the encoding of visual features emerged as early as 70 ms after stimulus onset in the VTC (Fig 2D). Surprisingly, semantic encoding in the MTL reached significance at 90 ms (Fig 2H), while memorability encoding in the PFC began at 170 ms (Fig 2L). Furthermore, the encoding of visual features peaked at 350 ms in the VTC, followed by semantic encoding in the MTL at 410 ms, and memorability encoding in the PFC at 410 ms. This increasing latency from visual to semantic to memorability encoding suggests a temporal progression of object-related processing across a distributed neural network.

Together, these results offer a detailed spatiotemporal characterization of how three key attributes are encoded during visual object processing. Our findings reveal that although visual, semantic, and memorability representations may share overlapping neural substrates in certain brain areas—particularly the VTC—they are primarily encoded in distinct brain regions with different temporal dynamics.

## Dissecting visual, semantic, and memorability representations in the VTC

We observed neural representations of visual, semantic, and memorability features in the VTC, consistent with findings from recent neuroimaging studies [14,26,40]. A key question is how these distinct aspects of neural object representations relate to one another within the VTC. Leveraging the high spatiotemporal resolution of intracranial recordings, we further examined the spatial distribution and temporal dynamics of channels selective for each attribute within the VTC. Remarkably, channels selective for visual, semantic, and memorability features were largely independent (Fig 3A and 3B). The number of overlapping channels was not significant for any pair of attributes (Fig 3A; $\chi^2$-test: visual and semantic: $P = 0.42$; visual and memorability: $P = 0.33$; semantic and memorability: $P = 0.65$). Visualization of individual channels in a 3D representational space further confirmed distinct clusters corresponding to each attribute (Fig 3B; see Fig 3C for temporal dynamics). To further validate the unique contributions of each attribute to object coding in the VTC, we employed partial correlation analyses (see Methods) within the RSA framework and revealed consistent results (Fig 3D; all $P$s $< 0.001$).

Interestingly, population-level RSA (Fig 3E) revealed that visual representation contributed to memorability representation (Spearman's $\rho = 0.082$, permutation $P = 0.0022$). However, there was no significant relationship between visual and semantic representations (Spearman's $\rho = 0.01$, permutation $P = 0.36$) nor between semantic and memorability representations (Spearman's $\rho = 0.03$, permutation $P = 0.11$), consistent with the results that the representations of visual, semantic, and memorability features involved distinct neural populations within the VTC.

Together, although the VTC is engaged in encoding visual, semantic, and memorability attributes, these attributes are dissociable.

## Interaction between memorability encoding and visual and semantic processing

We observed that memorability features were most robustly represented in the PFC, consistent with the hypothesis that the frontal cortex contributes to memory formation [46]. However, the mechanisms by which memorability representations arise in the frontal areas during visual object processing remain largely unclear. Recent evidence suggests

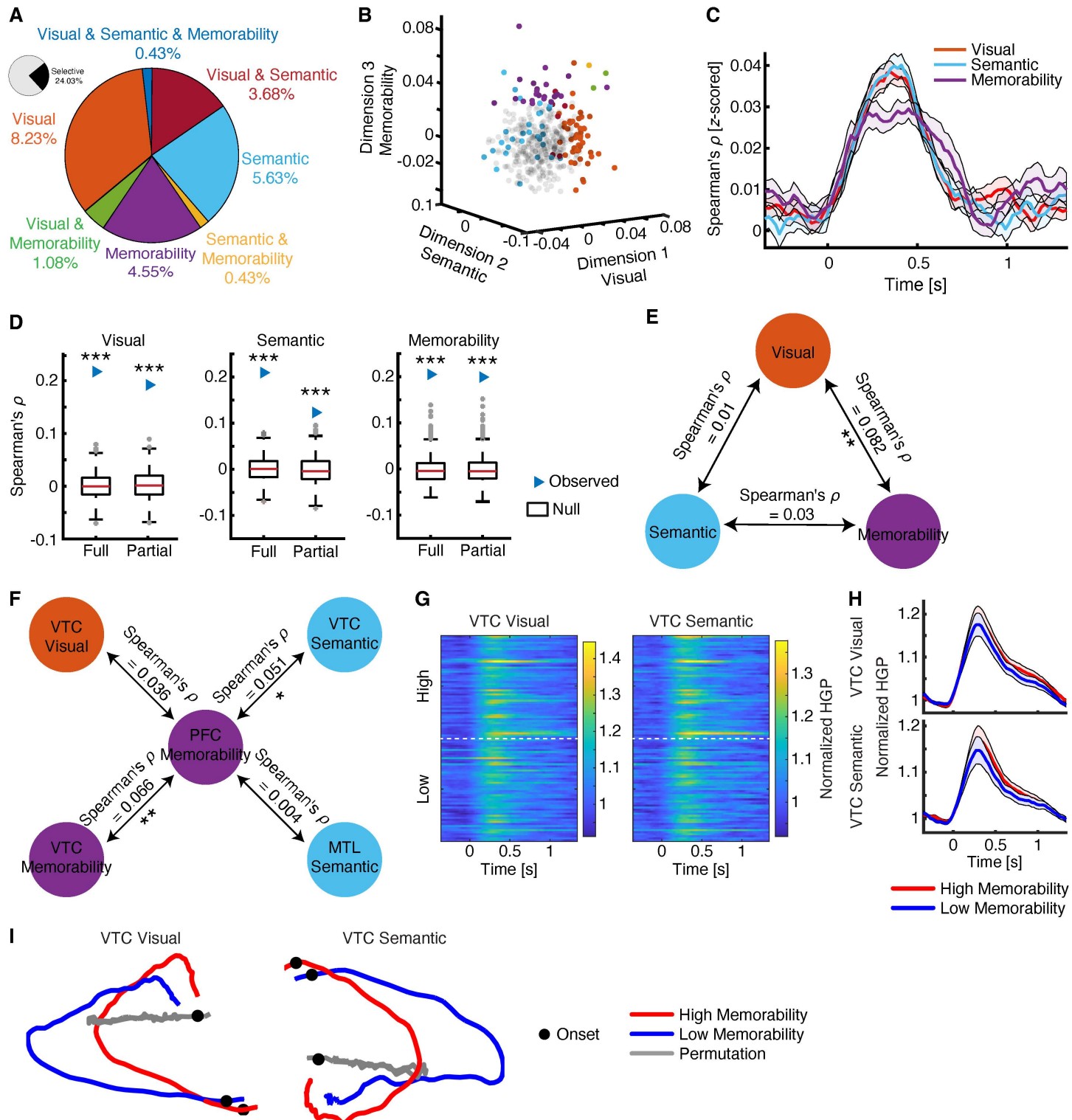

**Fig 3. Distinct neural populations in the VTC represent the visual, semantic, and memorability spaces. (A)** Percentage of channels representing each attribute (visual, semantic, and memorability). Note that only the selective channels (black in the inset) are shown. **(B)** Clustering of individual channels encoding visual, semantic, and memorability features. The dimensions represent Spearman's correlation coefficient between each channel's neural RDM and the corresponding feature RDMs. Each dot represents a channel. Color coding follows panel (A) and indicates the attribute encoded by each channel. Gray: nonselective channels. **(C)** Temporal dynamics of visual, semantic, and memorability representations. We only included channels

encoding a single attribute. **(D)** Group average of Pearson's correlation coefficients using full and partial correlations. Unique variance explained by each attribute was calculated using partial correlation while controlling for the other two feature types. On each box, the central mark is the median, the edges of the box are the 25th and 75th percentiles, the whiskers extend to the most extreme data points the algorithm considers to be not outliers, and the outliers are plotted individually. Asterisks indicate statistical significance based on a permutation test, comparing the observed $r$ (blue arrow) with the null distribution estimated from 1,000 permutations. ***: $P < 0.001$. **(E)** Representational similarity analysis (RSA) between neural populations encoding a single attribute within the VTC. **(F)** RSA of visual, semantic, and memorability encoding across brain areas. Asterisks indicate a significant difference using a permutation test. *: $P < 0.05$ and **: $P < 0.01$. **(G)** Normalized high-gamma power (HGP) for object images with high (top 50) vs. low (bottom 50) memorability scores. Each row represents an image, sorted by memorability score. Color coding indicates HGP averaged across visual-coding (left) or semantic-coding (right) channels in the VTC. Time 0 denotes image onset. **(H)** Mean normalized HGP for images with high vs. low memorability scores. Error bars denote ±SEM across channels. **(I)** Population dynamics of visual (left) and semantic (right) representations in the VTC for object images with high vs. low memorability scores. The source data underlying this figure are provided in S3 Data.

that semantic features, rather than visual perceptual features, play a larger role in predicting memorability scores [23]. Yet, the neural mechanisms underlying how visual and semantic processing influence memorability remain unknown. To address these questions, we investigated the information flow across brain areas predominantly involved in visual, semantic, and memorability processing. Specifically, we conducted RSA between PFC-memorability channels and potential sources of visual and semantic information from upstream brain areas, including VTC-visual channels, VTC-semantic channels, VTC-memorability channels, and MTL-semantic channels. We found that the memorability-encoding (Spearman's $\rho = 0.066$, $P = 0.003$) and semantic-encoding (Spearman's $\rho = 0.051$, $P = 0.018$) channels in the VTC shared a similar representational structure with the PFC-memorability channels (Fig 3F), suggesting that memorability representations in the PFC may arise from these VTC channels, which integrate visual and semantic information from other neural populations within the VTC.

To further explore how memorability representation interacts with visual and semantic processing, we analyzed the population dynamics (see Methods) for the top 50 most memorable images versus the bottom 50 least memorable images (see S1C Fig for the distribution of memorability scores) within the neural populations representing visual and semantic features. The representational geometry of object images with different memorability scores revealed distinct trajectories following stimulus onset for both visual (Fig 3I, left) and semantic representations (Fig 3I, right). The divergence of visual and semantic representations for high- versus low-memorability images in neural state space suggests that memorability modulates the encoding of object features during early perceptual and semantic processing stages. Importantly, in contrast to the representational geometry, the overall neural response was similar across high- versus low-memorability images (Fig 3G and 3H). This suggests that representational geometry captures additional information beyond that conveyed by population-averaged neural responses.

Together, our results suggest that memorability representations in the PFC are shaped by integrated visual and semantic information from the VTC, and that memorability modulates early stages of visual and semantic processing during object perception.

### Validation and generalization with high-resolution 7T fMRI data

To examine how the neural object representations revealed by the current iEEG study align with recent neuroimaging findings [26], we utilized a well-established high-resolution 7T fMRI dataset in which healthy participants viewed a large set of natural scene stimuli [47]. We included 2000 images from 20 object categories, with the number of images balanced across categories (see [48] for analysis from our previous study). Specifically, we implemented a partial least squares (PLS) regression model to test the encoding of visual and semantic features in the FG and MTL, which exhibited the highest percentage of channels representing visual and semantic features, respectively, based on iEEG activity.

First, we observed robust encoding of both visual (Fig 4A) and semantic (Fig 4F) features in the FG, consistent with previous neuroimaging results [40,49,50]. We also observed encoding of both visual (Fig 4A) and semantic (Fig 4F)

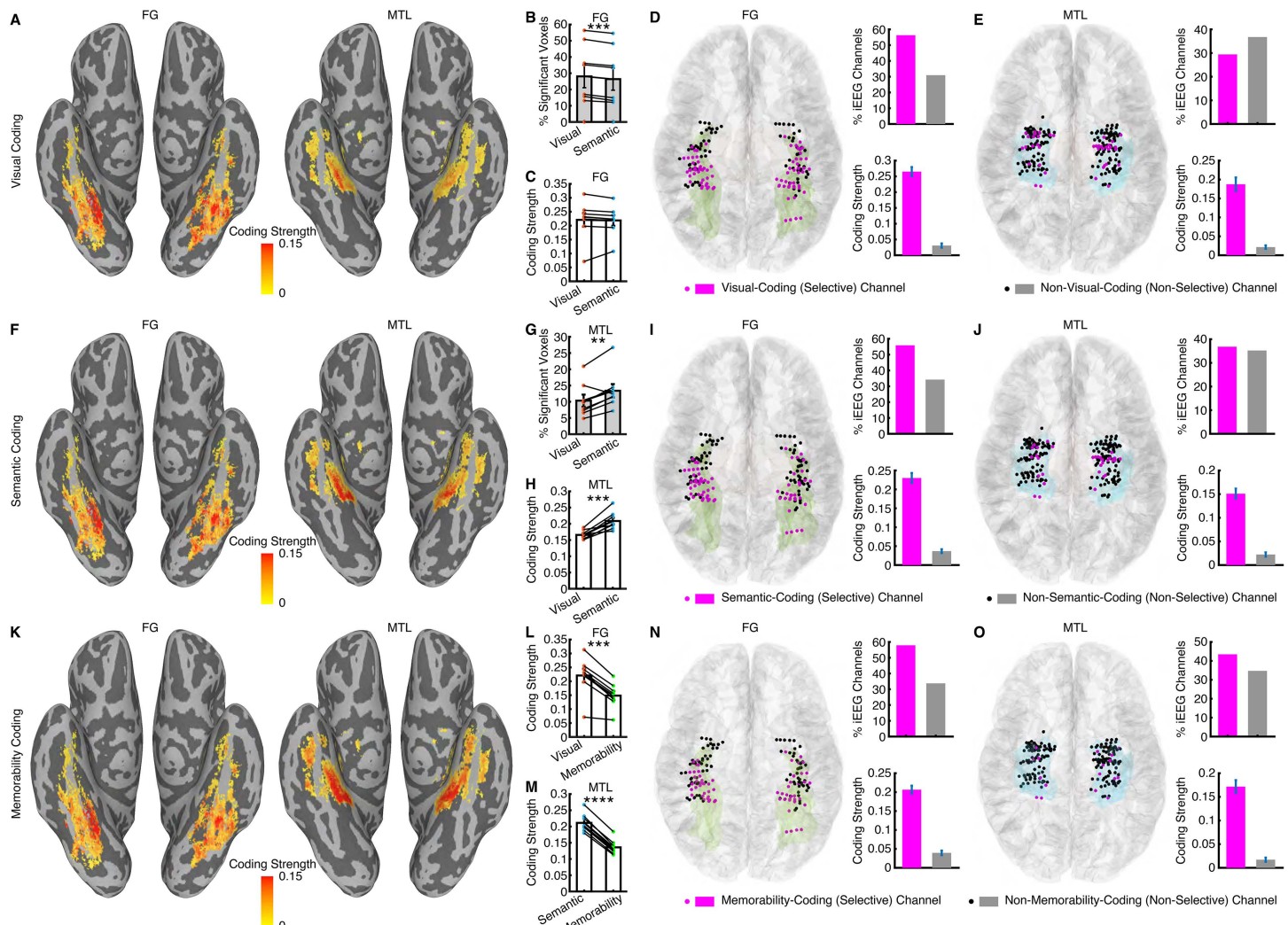

**Fig 4. Validation and generalization with high-resolution 7T fMRI data. (A–C, F–H, K–M)** Encoding of visual, semantic, and memorability features using 7T fMRI data. **(A)** Visual coding. **(F)** Semantic coding. **(K)** Memorability coding. Color coding indicates model coding strength, estimated by the Pearson's correlation between the observed and predicted responses. (left) Fusiform gyrus (FG). (right) Medial temporal lobe (MTL). **(B, C, G, H)** Comparison between visual and semantic coding. **(L)** Comparison between visual and memorability coding. **(M)** Comparison between semantic and memorability coding. **(B, G)** Percentage of significant voxels encoding visual or semantic models. **(C, H, L, M)** Average model coding strength for visual-coding, semantic-coding, or memorability-coding voxels. Each dot represents a single participant, and error bars denote ±SEM across participants (n = 8). Asterisks indicate a significant difference using a two-tailed paired t test across participants. **: P < 0.01, ***: P < 0.001, and ****: P < 0.0001. **(D, E, I, J, N, O)** Alignment of the fMRI and iEEG results. **(D, E)** Visual coding. **(I, J)** Semantic coding. **(N, O)** Memorability coding. **(D, I, N)** FG. **(E, J, O)** MTL. (left) Spatial alignment of iEEG channels with significant fMRI voxels encoding visual, semantic, or memorability features. (top right) Percentage of iEEG channels overlapping with significant fMRI voxels among selective (i.e., visual-coding, semantic-coding, or memorability-coding; magenta) vs. non-selective (non-visual-coding, non-semantic-coding, or non-memorability-coding; gray) channels. (bottom right) Model coding strength of iEEG channels overlapping with significant fMRI voxels, comparing selective vs. nonselective channels. The source data underlying this figure are provided in S4 Data.

features in the MTL, consistent with our previous study [48]. Notably, in line with our iEEG findings, the FG contained more significant voxels encoding the visual model than the semantic model (Fig 4B; two-tailed paired t test: t(7) = 5.61, P = 0.0004), whereas the MTL contained more significant voxels encoding the semantic model than the visual model (Fig 4G; t(7) = 3.43, P = 0.0055). These results were further supported by greater model coding strength for semantic

features in the MTL (Fig 4H; $t(7) = 5.23$, $P = 6.09 \times 10^{-4}$; for visual features in the FG: $t(7) = 0.37$, $P = 0.36$; Fig 4C). In addition, although we also observed robust encoding of memorability features in both the FG and MTL (Fig 4K), model coding strength was significantly greater for visual features in the FG (Fig 4L; $t(7) = 7.55$, $P = 1.31 \times 10^{-4}$) and for semantic features in the MTL (Fig 4M; $t(7) = 28.23$, $P = 1.79 \times 10^{-8}$), consistent with our iEEG results above.

We next examined the spatial alignment of fMRI and iEEG results. For visual features (Fig 4D and 4E), the percentage of visual-coding iEEG channels that overlapped with visual-coding fMRI voxels was significantly higher than that of non-visual-coding channels in the FG ($x^2$-test: $P = 0.0019$), but not in the MTL ($P = 0.42$). For semantic features (Fig 4I and 4J), the percentage of semantic-coding iEEG channels that overlapped with semantic-coding fMRI voxels was also higher than that of non-semantic-coding channels in the FG ($P = 0.0094$; but not in the MTL: $P = 0.87$). Similarly, for memorability features (Fig 4N and 4O), the percentage of memorability-coding iEEG channels that overlapped with memorability-coding fMRI voxels was higher than that of non-memorability-coding channels in the FG ($P = 0.0042$; but not in the MTL: $P = 0.49$). As expected, we further confirmed greater model coding strength in the fMRI-overlapping iEEG channels that significantly encoded visual (Fig 4D and 4E; two-tailed two-sample $t$ test: FG: $t(108) = 12.44$, $P < 10^{-20}$; MTL: $t(99) = 12.49$, $P < 10^{-20}$), semantic (Fig 4I and 4J; FG: $t(108) = 13.73$, $P < 10^{-20}$; MTL: $t(99) = 10.53$, $P < 10^{-20}$), and memorability (Fig 4N and 4O; FG: $t(108) = 13.80$, $P < 10^{-20}$; MTL: $t(99) = 10.96$, $P < 10^{-20}$) models, compared to those fMRI-overlapping iEEG channels that did not encode these models.

Together, consistent with the iEEG findings, high-resolution 7T fMRI further validated the encoding of visual and semantic features in the VTC and MTL using a different imaging modality.

## Validation and generalization with an additional dataset

We further validated our findings using an additional dataset comprising Microsoft COCO images [51]. This dataset included 10 common object categories, with 50 images per category, allowing for a more detailed examination of neural representations within categories. Consistent with our earlier results, we confirmed the encoding of visual features in the VTC ($n = 33$, 12.27%, binomial $P = 8.15 \times 10^{-7}$) and MTG ($n = 5$, 9.62%, binomial $P = 0.04$; Fig 5A and 5B); semantic features in the MTL ($n = 17$, 10.24%, binomial $P = 0.0018$) and VTC ($n = 25$, 9.29%, binomial $P = 0.0011$; Fig 5E and 5F); and memorability features in the VTC ($n = 29$, 10.78%, binomial $P = 4.02 \times 10^{-5}$), PFC ($n = 7$, 9.33%, binomial $P = 0.033$), and MTL ($n = 15$, 9.04%, binomial $P = 0.0094$; Fig 5I and 5J). Furthermore, the VTC, MTL, and PFC represented the corresponding visual (Fig 5C; Spearman's $\rho = 0.17$, permutation $P < 0.001$), semantic (Fig 5G; Spearman's $\rho = 0.15$, permutation $P < 0.001$), and memorability (Fig 5K; Spearman's $\rho = 0.08$, permutation $P < 0.001$) features, respectively, at the population level. Similarly, visual encoding in the VTC peaked first (Fig 5D; 290 ms), followed by semantic encoding in the MTL (Fig 5H; 450 ms), and memorability encoding in the PFC (Fig 5L; 450 ms). Therefore, these findings demonstrated the generalizability of our results across different datasets and confirmed the dissociation and distinct neural dynamics of visual, semantic, and memorability representations across brain regions.

## Encoding of visual, semantic, and memorability features at the single-neuron level

Human single-neuron recordings have provided valuable insights into the mechanisms underlying memory [52,53]. Individual neurons encoding specific concepts are considered the building blocks of declarative memory [54]. But how do the results observed at the mesoscopic and macroscopic scales reflect underlying neuronal activity? And how are object features encoded at the level of individual neurons? To address these questions, we recorded from 1,770 neurons across 15 neurosurgical patients (9 females) in the amygdala and hippocampus. Of these, 1,237 neurons (28 sessions from 15 patients) were recorded during the ImageNet task (see representative neurons in Fig 6A and 6B) and 533 neurons (18 sessions from 13 patients) were recorded during the COCO task (see representative neurons in Fig 6C and 6D).

We examined the representation of visual, semantic, and memorability features using the RDM model, as in the iEEG analysis. The results were qualitatively consistent with the iEEG findings: a significant proportion of single neurons in the

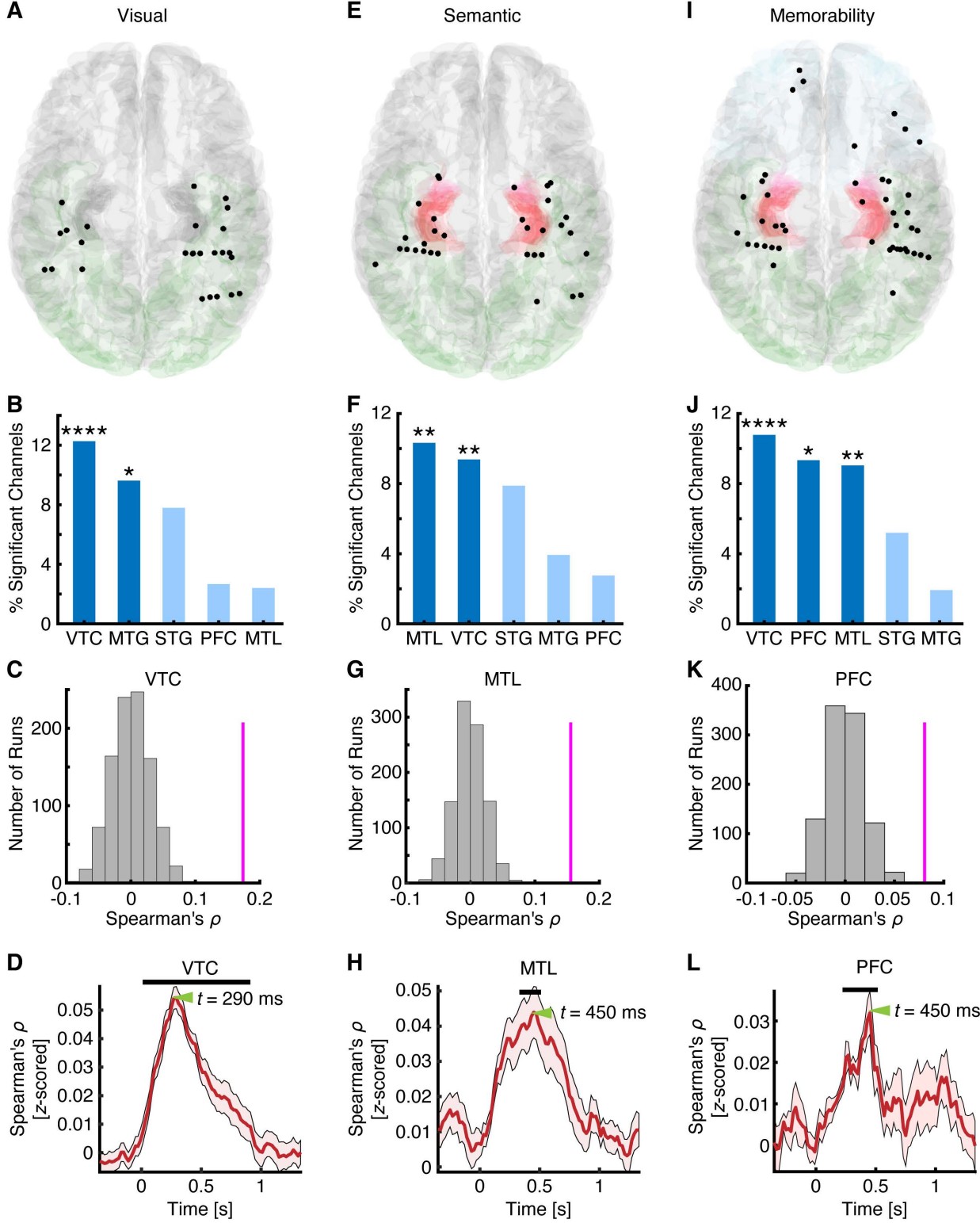

**Fig 5. Validation and generalization with the COCO dataset.** Legend conventions as in Fig 2. MTG: middle temporal gyrus. MTL: medial temporal lobe. PFC: prefrontal cortex. STG: superior temporal gyrus. VTC: ventral temporal cortex. The source data underlying this figure are provided in S5 Data.

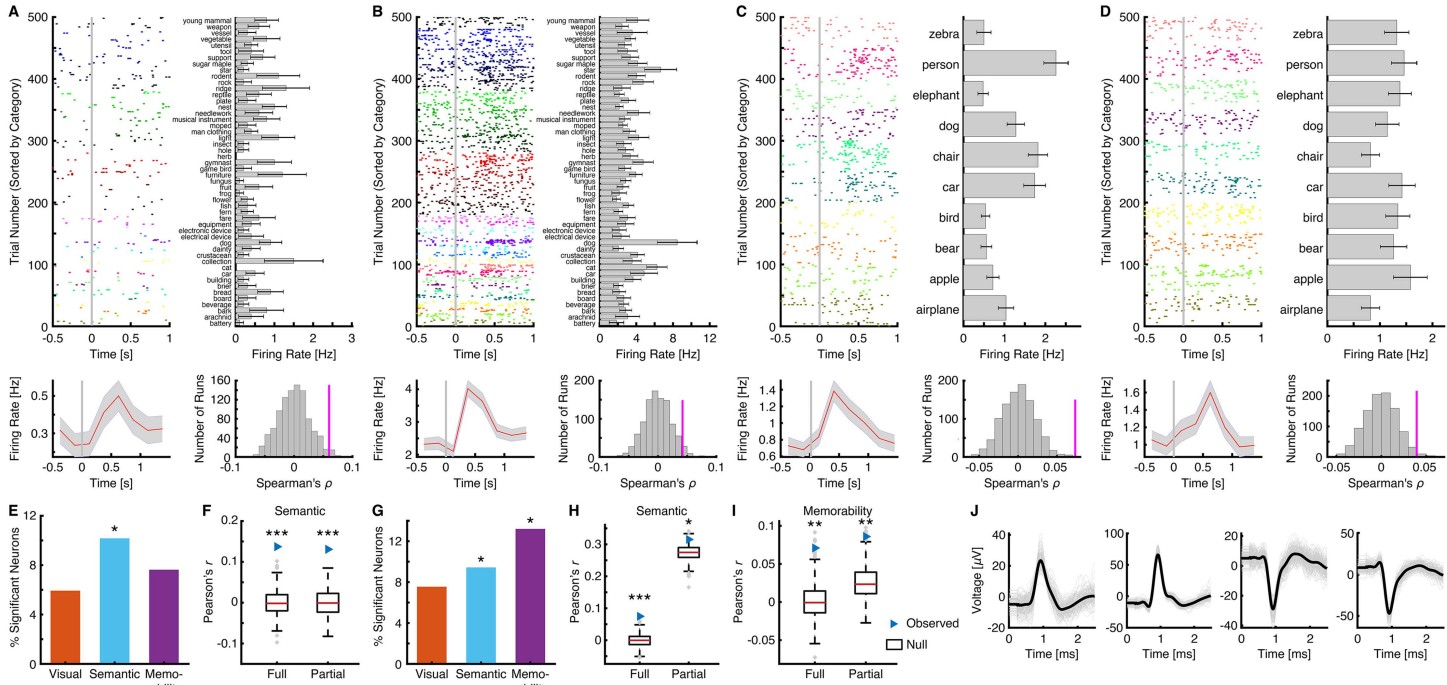

**Fig 6. Encoding of visual, semantic, and memorability features at the single-neuron level. (A–D)** Example MTL neurons encoding **(A–C)** semantic and **(D)** memorability features from the **(A, B)** ImageNet and **(C, D)** COCO tasks. Upper left: Spike firing in response to 500 objects (50 object categories for the ImageNet task and 10 categories for the COCO task). Trials are aligned to stimulus onset (gray line) and grouped by individual object category. Upper right: Firing rate for each category, averaged across trials. Error bars denote ±SEM across trials. Bottom left: Peristimulus time histogram (PSTH) estimated with a sliding window (0.25 s) from –0.5 to 1.5 s relative to stimulus onset. Shaded areas denote ±SEM across trials. Bottom right: Spearman correlation between the neural RDM of the example neuron and the **(A–C)** semantic or **(D)** memorability RDM. Magenta bars indicate the observed correlation coefficients, while gray bars represent the null distribution estimated via permutation (1,000 runs). Waveforms of the neurons are shown in **(J)**. **(E, G)** Percentage of neurons encoding visual, semantic, and memorability features in the MTL. Asterisks indicate a significant number of neurons, as determined by a binomial test. *: $P < 0.05$. **(F, H, I)** Group average of Pearson's correlation coefficients using full or partial correlation. **(E, F)** ImageNet task. **(G–I)** COCO task. Legend conventions as in Fig 3D. The source data underlying this figure are provided in S6 Data.

MTL encoded semantic features in both the ImageNet (Fig 6E; $n = 12$, 10.17%, binomial $P = 0.0063$; see example neurons in Fig 6A and 6B) and COCO (Fig 6G; $n = 5$, 9.43%, binomial $P = 0.048$; see an example neuron in Fig 6C) tasks. In contrast, we observed a significant percentage of neurons encoding memorability only during the COCO task (Fig 6G; $n = 7$, 13.21%, binomial $P = 0.0046$; see an example neuron in Fig 6D). These findings were further validated using correlation coefficients averaged across the whole neuronal population, using both full and partial correlations for the ImageNet task (Fig 6F; full: $r = 0.14$, permutation $P < 0.001$; partial: $r = 0.13$, permutation $P < 0.001$) and the COCO task (Fig 6H and 6I; semantic: full: $r = 0.07$, permutation $P < 0.001$; partial: $r = 0.32$, permutation $P = 0.037$; memorability: full: $r = 0.07$, permutation $P = 0.002$; partial: $r = 0.09$, permutation $P = 0.004$). Therefore, these results further corroborated our findings at the single-neuron level in the human MTL.

## Discussion

In this study, we comprehensively investigated how visual, semantic, and memorability attributes are dynamically encoded across the brain during visual object perception. By characterizing the spatiotemporal patterns of neural encoding using iEEG, we demonstrated that these attributes are encoded in distinct brain regions with different temporal dynamics. Our findings reveal that memorability representations in the PFC arise from integrated visual and semantic signals from the

VTC, and that image memorability modulates both visual and semantic representations in the VTC, supporting an interaction between perception and memory [55]. Our results were robustly validated using neural signals across different spatio-temporal scales, including high-resolution 7T fMRI across the brain and single-neuron recordings in the MTL, consistent with our previous analyses of visual object coding [17,48]. Additionally, we confirmed the generalizability of our findings using a different set of stimuli featuring within-category variability. Together, this study highlights an integrated neural computational pathway for visual perception [43,44]. The multimodal neural signals complemented one another, bridging insights from previous neuroimaging studies and current electrophysiological evidence. This study not only advances our understanding of the brain's complex processes involved in object recognition and memory formation but also provides a foundation for future studies investigating how perceptual and memory-related attributes interact to shape experience.

Recent advancements in deep learning have revolutionized the tools used to understand object vision in the human brain. This revolution is shifting the field's focus toward a multidimensional understanding of object representations that extend beyond visual features, incorporating information from other modalities conveyed by object images, such as semantics, context, and memorability [8,55,56]. This multidimensional framework is supported by the high generality and efficiency of vision-language models like CLIP [57], which are pre-trained on paired image-text data for object recognition. A recent study showed that embeddings derived from CLIP explained additional variance in human perception and memory of faces and objects compared to visual DNNs [38]. Furthermore, language-aligned visual models and large language models (LLMs) have been shown to align well with neural responses in the human visual cortex, as measured by fMRI [40,49,50]. These recent findings have sparked increased interest in the relative contributions of semantic and visual experiences to object recognition, as well as in the functional organization of the visual cortex [50].

In line with these developments, we demonstrated mesoscale- and microscale-level neural representations of semantic features derived from LLMs in the human VTC and MTL. Despite the spatial co-localization of visual and semantic representations in the VTC, these processes recruited distinct neural populations. Importantly, the MTL exhibited a stronger preference for semantic representation at both the single-neuron level and within local neural populations, forming an anterior-medial semantic topography (Fig 2C), whereas visual representation was predominantly localized to the posterior and lateral temporal cortex (Fig 2A). Moreover, the temporal dynamics of semantic representation were relatively delayed compared to those of visual representation. Together, our finer-grained spatiotemporal analysis revealed disentangled visual and semantic representations in the higher visual cortex, extending recent neuroimaging findings that primarily focused on posterior visual areas [40,49,50,58].

Extending beyond sensory perception, the current study investigated how visual and semantic representations relate to memory by examining the intrinsic object memorability. This inquiry was motivated by recent computational modeling studies revealing interactions among visual, semantic, and memorability properties in object images [8,23,55,59,60]. By aligning the latent space of the memorability-predicting ResMem model [7] with iEEG activity, we demonstrated neural encoding of memorability in the VTC and MTL, echoing findings from recent fMRI studies [14,15,24]. Interestingly, we found that the PFC exhibited stronger encoding of memorability than both the visual cortex and the mnemonic MTL regions. The memorability signal emerged at a later latency than visual and semantic signals, supporting the idea that memorability processing builds upon prior visual and semantic processing. Furthermore, the VTC neural populations encoding memorability were distinct from those encoding visual and semantic features, even though their representations were mutually aligned (Fig 3). Notably, memorability representation in the PFC was most similar to that in the VTC—compared to VTC visual, VTC semantic, and MTL semantic representations—suggesting that the PFC accesses information from memorability-specific neural populations in the VTC, which may integrate visual and semantic information via separate neural circuits. These findings delineate a finer-grained pathway for the encoding of object memorability, establishing a neural link between object perception and memory.

The interaction between perception and memory is widely considered bidirectional [55], yet the mechanisms by which memory shapes visual and semantic perception remain elusive. A well-documented example of memory influencing

perception is the familiarity effect, observed at both neural [61–64] and behavioral [65,66] levels, suggesting that memory enhances perceptual processing. Our recent study further demonstrated that high memorability enhances feature-based object processing in the MTL [17]. Memorability has also been shown to influence the subjective perception of time [67]. In the present study, we found that memorability robustly modulated the population dynamics of object processing in both visual- and semantic-encoding neural populations within the VTC, highlighting memory's role in shaping visual and semantic representations.

Our findings have important implications for evaluating how well current artificial intelligence models—such as DNNs and LLMs—capture human visual, semantic, and memorability representations. While DNNs trained on object recognition tasks often mirror aspects of visual feature coding in the VTC [41–43], they typically lack representations that correspond to semantic integration and memory-related processes observed in the MTL and PFC. LLMs trained with visual descriptions can capture both semantic relationships based on language structure and visual experiences, as shown in recent publications [49,50]. However, neither the visual DNN models and LLMs inherently model perceptual memorability and they were mostly studied in separate studies. In the present study, our results suggest that human memorability representation arises from the interaction of perceptual and semantic systems and subsequently feeds back to modulate perceptual encoding. This integrative process is not yet well-captured by current AI models, pointing to the need for future models that incorporate multimodal learning and memory mechanisms to better reflect human cognitive architecture.

Despite the advantages of iEEG in spatiotemporal resolution and high signal-to-noise ratio (SNR), we acknowledge that our recordings lacked coverage of the posterior temporal cortex. To address this limitation, we conducted parallel analyses using the NSD dataset [47], collected with a 7T MRI scanner and widely used in recent studies of visual object perception [48,49,58]. Our results demonstrated broadly aligned visual and semantic representations across the posterior to middle FG and MTL, consistent with previous studies [40,49,58]. Notably, we observed stronger visual representations in the VTC and stronger semantic representations in the MTL, complementing and confirming our iEEG findings. Furthermore, we directly validated our results at the single-neuron level using two independent datasets in the MTL, providing robust neuronal correlates of semantic and memorability representations. These multimodal findings not only reinforce the conclusions of the current study but also underscore the power of integrating methodologies to achieve a comprehensive understanding of human perception and memory.

In this study, we did not implement a memory task, and hence no direct predictor of individual memory was used to model neural responses. However, in our previous study, which included a memory task, we showed that memorability scores estimated from this DNN are highly correlated with human memory performance [17]. Furthermore, the relationship between memorability scores or features and memory performance has been validated in numerous prior studies [6–8,60,68]. These previous studies support our current findings using predicted memorability scores. Furthermore, in the present study, we defined the MTL as encompassing both the hippocampus and amygdala. While these structures may contribute differently to memory processes, our previous study using the same object stimuli [17] found that both regions exhibited similar response patterns with respect to memory and memorability. Based on this similarity, and to maximize statistical power, we combined them into a single MTL ROI. Nonetheless, we acknowledge that the hippocampus and amygdala may support distinct aspects of memory processing, and future work with larger datasets will be necessary to dissociate their respective contributions to perceptual memorability.

We averaged neural responses across repetitions of the same image to improve SNR. While repetition can, in principle, influence neural responses due to familiarity or adaptation effects, prior analyses of the NSD dataset have shown that response patterns remain relatively stable across multiple presentations of the same image [47]. Moreover, memorability is considered an intrinsic property of images that evokes consistent neural responses across individuals and viewing experiences [7]. As such, repetition is unlikely to fundamentally alter the encoding of memorability itself, but may instead influence later memory retrieval processes. Thus, averaging across repetitions allows us to capture a more robust measure of the stimulus-driven neural signal without significantly diluting the memorability-related response.

In conclusion, the dissociable yet interconnected neural representations of visual, semantic, and memorability features in object perception support an emerging multidimensional theory of object recognition [56]. That is, processing different attributes embedded in object images—such as visual, semantic, and memorability—involves a distributed network, including those traditionally associated with visual or semantic processing. Specifically, these processes interact early in the visual cortex, enabling efficient object representations that extend beyond categorical distinctions to include semantic relationships and memory. Our study initiates a coherent, neural-level investigation of perception and memory within a multidimensional processing framework. Future research should integrate task-based designs, behavioral data, computational modeling, and neural recordings to further elucidate the complex nature of object recognition.

## Methods

### Participants

We recorded iEEG and/or single-neuron activity from neurosurgical patients with drug-resistant epilepsy. All participants provided written informed consent under procedures approved by the Institutional Review Boards of Washington University in St. Louis (WUSTL; protocol number: 202201019) and West Virginia University (WVU; protocol number: 1709745061). The study was conducted in accordance with the principles of the Declaration of Helsinki.

For the ImageNet task, we recorded iEEG from 20 patients (12 females; 30 sessions in total; S1 Table) while presenting object images from the ImageNet dataset [29]. In 13 of these sessions, we simultaneously recorded single-neuron activity along with iEEG. Additionally, we recorded only single-neuron activity in the amygdala and hippocampus from 7 additional patients (15 sessions).

For the Microsoft COCO task, we recorded iEEG from 11 patients (7 females; 16 sessions in total). In 13 of these sessions, we simultaneously recorded single-neuron activity along with iEEG. Furthermore, we recorded only single-neuron activity in the amygdala and hippocampus from 5 additional patients (5 sessions).

### Data acquisition and preprocessing

We recorded iEEG signals using stereotactic clinical depth-electrodes (PMT, DIXI, or Ad-Tech) connected to a Nihon Kohden recording system. Each electrode contained 8–16 channels depending on the trajectory length. Data were digitally sampled at 2,000 Hz and analog-filtered above 0.01 Hz during acquisition. Preprocessing was performed using EEGLAB [69]. Initially, we visually inspected the data and removed channels contaminated by epileptic or other artifactual activity (e.g., movement artifacts, line noise). We then applied a 0.5 Hz high-pass filter and a common average reference, followed by a notch filter to remove line noise (60 Hz and its harmonics at 120 and 180 Hz). Continuous data were segmented into event-related epochs spanning −0.5 to 1.5 s relative to stimulus onset. High-amplitude noise events and interictal discharges were identified on a trial-by-trial basis using an automatic thresholding procedure described in [70].

To derive responses of local neural populations, we extracted HGP by taking the absolute value of the Hilbert transform of the bandpass filtered (70–170 Hz) signal. For all analyses, except for the time-resolved RSA, we used the average response for each object image within the 0.1- to 0.6-second time window following stimulus onset. Additionally, we extracted the HGP time course for each channel to analyze the temporal profiles of neural responses.

Single-neuron signals were recorded using microwires embedded in hybrid depth-electrodes (Ad-Tech Behnke-Fried electrodes) implanted in the amygdala and hippocampus. Broadband extracellular signals (0.1–9,000 Hz) were recorded from each microwire at a sampling rate of 32 kHz. Data were continuously stored using either a Blackrock (WUSTL) or Neuralynx (WVU) system. For sessions with concurrent recordings, the iEEG and single-neuron signals were time-synchronized using a photodiode patch attached to the stimulus screen. The raw single-neuron data were filtered with a zero-phase lag bandpass filter (300–3,000 Hz), and spikes were sorted with a semi-automatic template matching algorithm (OSort) as described in [71]. During spike sorting, we manually removed clusters with poor waveforms that did

not resemble neuronal activity. For further analysis, we extracted each neuron's mean firing rate within the 0.25–1.25 s window following stimulus onset for each image.

The fMRI data were from the NSD dataset [47] that contained high-resolution BOLD-fMRI responses to thousands of images from Microsoft's COCO database from eight healthy participants (6 females). We used the single-trial betas of the upsampled 1.0-mm high-resolution preparation of the NSD data [47]. We only included images that were seen three times, with single-trial betas averaged across the 3 repetitions voxel-by-voxel. To ensure that each object category had a sufficient and balanced number of images, we only included the 20 object categories that had at least 100 images. For each object category, we randomly selected 100 different images for analysis. In total, 2,000 images were included for each participant. Although different participants viewed different sets of images, we used the same object categories across participants. The object categories include person, bear, bench, book, bowl, cat, cow, dog, giraffe, train, airplane, bed, bird, bottle, car, chair, clock, elephant, toilet, and zebra. It is worth noting that this random sampling strategy was chosen to avoid potential selection bias and to maintain the ecological validity of the stimulus set. Importantly, the natural variability present in the source dataset means that random sampling inherently captures a wide range of image features, ensuring broad and representative coverage of visual, semantic, and memorability variability within each object category (see [48] for detailed analysis). We verified that the selected stimuli spanned a wide range of visual, semantic, and memorability changes, supporting the generalizability of our results. Therefore, this approach allowed us to examine neural encoding in response to stimuli that reflect real-world variation, rather than restricting analyses to handpicked exemplars based on specific criteria.

## Electrode localization

Electrode localization was performed using the VERA software suite (https://github.com/neurotechcenter/VERA), which utilizes co-registered pre-operative high-resolution T1-weighted MRI scans and post-operative CT scans. Each channel was assigned 3-dimensional coordinates and an anatomical label derived from Freesurfer's automatic segmentation [72]. The coordinates were then normalized to the MNI space through nonlinear co-registration for visualization.

## Stimuli

Two sets of stimuli were used in the current study, and the same images were used for all participants.

For the ImageNet stimuli, we selected 50 object categories from the ImageNet dataset [29], with each category containing 10 unique images. The object categories included arachnid, battery, bark, beverage, board, bread, brier, building, car, cat, collection, crustacean, dainty, dog, electrical device, electronic device, equipment, fare, fern, fish, flower, frog, fruit, fungus, furniture, game bird, gymnast, herb, hole, insect, light, man clothing, moped, musical instrument, needlework, nest, plate, reptile, ridge, rock, rodent, star, sugar maple, support, tool, utensil, vegetable, vessel, weapon, and young mammal.

For the Microsoft COCO stimuli, we selected 10 object categories from the Microsoft COCO dataset [51], with each category containing 50 unique images. The categories included airplane, apple, bear, bird, car, chair, dog, elephant, person, and zebra.

## Experimental procedure

We displayed the object images in a randomized order on a flat monitor positioned in front of the patients, who sat comfortably on a bed in their patient room. The task began with a short fixation, after which the images were displayed sequentially. Each image was shown for a fixed duration of 1 s, followed by a uniformly jittered inter-stimulus interval ranging from 0.5 to 0.75 s. The images subtended a visual angle of approximately 10°. A simple one-back task was implemented during the presentation, where participants were instructed to press a button if the current image was identical to the immediately preceding one. One-back repetitions occurred in approximately 10% of trials. Each image was presented

once unless repeated during a one-back trial. Responses from one-back trials were excluded from the analysis to ensure an equal number of responses for each image. Notably, this task has been shown to effectively maintain patients' attention on the images while minimizing biases from focusing on specific image features [73].

### Visually responsive iEEG channels

We used the mean HGP within the time window of 0.1 to 0.6 s after stimulus onset as the neural response to each stimulus. A channel was considered visually responsive if it exhibited a significantly different response compared to the baseline period (0.5 to 0.02 s before stimulus onset). All analyses were restricted to stimulus-responsive channels.

### Construction of the visual, semantic, and memorability RDMs

We constructed RDMs for visual, semantic, and memorability features using well-established and advanced models. For visual features, we used the pre-trained ResNet [35] model for ImageNet stimuli and the AlexNet [74] model for COCO stimuli. Both models have demonstrated strong performance in object categorization for these datasets and have been shown to effectively model neural responses [17,42,47]. Following established procedures [75,76], we fine-tuned the top layer of each model to verify that the pre-trained models could effectively discriminate object categories, thereby validating their suitability as feature extractors. For semantic features, we first used the ALBEF model [77] to generate text descriptions for each image. These descriptions were then processed using the SGPT model [36] to extract sentence embeddings. SGPT was pre-trained for image-text alignment and has been shown to capture semantically rich representations that generalize across a wide range of cognitive and perceptual tasks, including aspects of memory [36]. For memorability features, we employed the ResMem model [37], which was trained and evaluated directly on human memorability ratings across large image sets and participant populations [7,37], to extract features predictive of image memorability.

In selecting feature representations from each model, we used the output layer as the feature extraction layer, with the exception of the AlexNet model, where we employed the FC6 layer in line with prior literature [42]. This choice was guided by the rationale that output layers contain the most task-relevant and abstracted representations, reflecting the culmination of computations optimized for specific behavioral or semantic objectives. For instance, in object classification models such as ResNet and AlexNet, the output layers encode high-level categorical or semantic structure, while in the ResMem model, the output is explicitly trained to predict human memory performance. Consequently, these layers are well-suited for assessing the alignment between model-derived features and neural activity associated with related cognitive functions, particularly memory encoding. Importantly, we confirmed that our main findings were robust across alternative higher-level layers for each feature type (S2A–S2E Fig), supporting the generalizability of our results beyond specific layer choices.

To visualize the geometric structures of the three sets of features, we applied t-Distributed Stochastic Neighbor Embedding, a dimensionality reduction technique designed for visualizing high-dimensional data.

### Representational similarity analysis (RSA)

We performed RSA [78] to map the visual, semantic, and memorability representations onto the brain and to compare neural representations across different brain areas. For a given channel or ROI, we constructed a neural RDM by calculating the pairwise Euclidean distances between object stimuli based on their neural responses. Specifically, for individual channels (Fig 2B, 2F, and 2J), we used the mean HGP (a single value) within a 0.1–0.6 s window after stimulus onset to compute Euclidean distances (i.e., absolute differences) for the RDMs. For population analysis in ROIs with multiple channels (Fig 2C, 2G, and 2K), we computed the Euclidean distances between neural vectors for the RDMs. To map visual, semantic, and memorability representations, the neural RDM of a given channel or ROI was correlated with each feature RDM using Spearman correlation, which does not assume a linear relationship. To compare representations between

ROIs, we computed Spearman correlations between their neural RDMs using the upper triangular portion of each matrix. Statistical significance was assessed using a Mantel test [79], which generated a null distribution by permuting the rows and columns of one RDM 1,000 times prior to correlation.

To assess the unique contributions of each visual, semantic, and memorability representation to neural activity, we computed partial correlations between each feature RDM and the neural RDMs, holding the other two feature RDMs constant.

We also conducted time-resolved RSA by correlating the neural RDMs estimated along the time series of each selected channel with each feature RDM. For computational efficiency, the raw time series were binned using a sliding window of 300 ms with a 280 ms overlap. At each time point, we computed the correlation between the neural RDM and each feature RDM. Statistical significance at each time point was determined using a paired $t$ test against baseline mean (within −0.5 to 0s relative to the stimulus onset; $P < 0.05$, FDR-corrected) across channels. We retained only consecutive significant data points lasting longer than 5 bins (100ms).

### Control analysis for individual differences

To rule out the possibility that our results were driven by specific participants, we performed a participant-level bootstrapping test and applied a GLME model, treating participant as a random effect, for the selection of channels representing visual, semantic, and memorability features. In the bootstrapping test (1,000 iterations), we randomly subsampled PFC channels from 15 patients (75% of the total sample) and calculated the percentage of selective channels in each iteration. If the observed percentage of selective channels fell within the bootstrapped distribution, this would suggest that the effect was not solely driven by individual participants. In the GLME analysis, we fit a full linear model with participant included as a random effect (percentage ~ 1 + (1 | participant)) and compared it to a reduced model without the random effect (percentage ~ 1). We then conducted a likelihood ratio test to determine whether including the random effect significantly improved model fit.

### Linear encoding models

We fitted a linear regression model to each channel or voxel to test whether neural representations of visual, semantic, and memorability features could be better explained by a linear model: $\boldsymbol{R = X\beta + \varepsilon}$, where $\boldsymbol{R}$ is the neural response vector, $\boldsymbol{X}$ is the feature matrix (visual, semantic, or memorability features), $\boldsymbol{\beta}$ is the weight vector, and $\boldsymbol{\varepsilon}$ is the error term. In particular, for memorability scores, we used simple linear regression (i.e., correlation) with the "fitlm" function in MATLAB, as the memorability space varied primarily along a single dimension reflecting the memorability score (Fig 1D, right). For high-dimensional visual, semantic, and memorability features (see above), we applied PLS regression using the "plsregress" function in MATLAB, following the procedure detailed in [43,44].

We used a permutation test with 1,000 runs to determine whether a channel or voxel encoded a significant model (i.e., the voxel encoded the dimensions of the feature spaces). In each run, we randomly shuffled the object labels and used 50% of the objects as the training dataset. We used the training dataset to construct a model (i.e., deriving regression coefficients), predicted responses using this model for each object in the remaining 50% of objects (i.e., test dataset), and computed the Pearson correlation between the predicted and actual response in the test dataset. The distribution of correlation coefficients computed *with* shuffling (i.e., null distribution) was eventually compared to the one *without* shuffling (i.e., observed response). If the correlation coefficient of the observed response was greater than 95% of the correlation coefficients from the null distribution, this model was considered *significant*. This procedure has been shown to be very effective to select neurons with significant face models [80]. The correlation coefficient thus could indicate the model's coding strength.

It is worth noting that we did not combine the three types of features into a single linear model, in order to avoid confounding effects due to feature collinearity and differing dimensionalities.

## Population dynamics

To investigate how memorability influences visual and semantic representations, we examined the average population dynamics of these representations for images with high versus low memorability scores (Fig 3G). Images with the top 50 and bottom 50 memorability scores were defined as high- and low-memorable, respectively. Raw time series data were binned using a sliding window of 200 ms with a step size of 20 ms, and then averaged separately for high- and low-memorable images. iEEG channels representing visual and semantic features were pooled separately, and principal component analysis (PCA) was applied to obtain state-space trajectories for the high- versus low-memorability conditions. The first three principal components were used for visualization. A null distribution was estimated by shuffling the high and low memorability labels 1,000 times prior to PCA. All analyses and visualizations were conducted using MATLAB.

## Supporting information

**S1 Fig. Stimuli, category selectivity, and neural RDMs. (A)** Object category eliciting the maximal response for each visually responsive channel. Each color represents a different object category. **(B)** Neural RDMs of population responses for each ROI, sorted by object category. The RDMs were computed using all visually responsive channels within each ROI. **(C)** Distribution of memorability scores for the ImageNet stimuli. **(D)** Distribution of memorability scores for the COCO stimuli. The source data underlying this figure are provided in S7 Data.
(PDF)

**S2 Fig. Control analyses for the encoding of visual, semantic, and memorability features. (A–E)** Percentage of significant channels as a function of model layer, illustrated here using the ventral temporal cortex (VTC). **(A)** Encoding of visual features using the ResNet-101 model. **(B)** Encoding of visual features using the VGG model. **(C)** Encoding of semantic features using the SGPT model. **(D)** Encoding of semantic features using the SBERT model. **(E)** Encoding of memorability features using the ResMem model. **(F)** Percentage of significant channels in each region of interest (ROI) using the MemNet model. **(G–I)** Population-level neural encoding of **(G)** visual, **(H)** semantic, and **(I)** memorability features in the most prominent ROI for each attribute. RDMs were constructed using cosine distance. **(J–L)** Percentage of significant channels encoding **(J)** visual, **(K)** semantic, and **(L)** memorability features using linear models. **(M–O)** Percentage of significant channels encoding **(M)** visual, **(N)** semantic, and **(O)** memorability features using data from the first session of each participant. Legend conventions as in Fig 2. The source data underlying this figure are provided in S8 Data.
(PDF)

**S1 Table. Patients and sessions for iEEG recordings of the ImageNet stimuli.** VTC, ventral temporal cortex; MTL, medial temporal lobe; PFC, prefrontal cortex; FG, fusiform gyrus; ITG, inferior temporal gyrus; Lingual, lingual gyrus; LOC, lateral occipital cortex; Amy, amygdala; AH, anterior hippocampus; PH, posterior hippocampus; ERC, entorhinal cortex; PHC, parahippocampal cortex; SFG, superior frontal gyrus; POp, pars opercularis cortex; MFG, middle frontal gyrus.
(PDF)

**S1 Data. Data for Fig 1.**
(XLSX)

**S2 Data. Data for Fig 2.**
(XLSX)

**S3 Data. Data for Fig 3.**
(XLSX)

**S4 Data. Data for Fig 4.**
(XLSX)

**S5 Data. Data for Fig 5.**
(XLSX)

**S6 Data. Data for Fig 6.**
(XLSX)

**S7 Data. Data for S1 Fig.**
(XLSX)

**S8 Data. Data for S2 Fig.**
(XLSX)

## Acknowledgments
We thank all patients for their participation.

## Author contributions
**Conceptualization:** Yue Wang, Runnan Cao, Shuo Wang.

**Data curation:** Peter Brunner, Jon T. Willie, Runnan Cao.

**Formal analysis:** Yue Wang, Runnan Cao.

**Resources:** Peter Brunner, Jon T. Willie.

**Software:** Peter Brunner.

**Supervision:** Shuo Wang.

**Writing – original draft:** Yue Wang, Runnan Cao, Shuo Wang.

**Writing – review & editing:** Yue Wang, Runnan Cao, Shuo Wang.

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
