## [Editor Report · Decision Letter 0]

5 Sep 2025

Dear Shuo,

Thank you for submitting your manuscript entitled "Neural computations of visual, semantic, and memorability features in the human brain" for consideration as a Research Article by PLOS Biology. Please allow me to apologize for the long delay in getting back to you. It took us a bit longer than usual to discuss your appeal with an Academic Editor.

In any case, your manuscript has now been evaluated by the PLOS Biology editorial staff as well as by an academic editor with relevant expertise and I am writing to let you know that we would like to send your submission back for external peer review. I should say that we are not fully convinced that all reviewer concerns are sufficiently addressed and the revision leaves a sufficiently strong manuscript for PLOS Biology, but we would like to see what the reviewers think about the revision. We will be looking for strong support from the reviewers.

Once your full submission is complete, your paper will undergo a series of checks in preparation for peer review. After your manuscript has passed the checks it will be sent out for review. To provide the metadata for your submission, please Login to Editorial Manager (https://www.editorialmanager.com/pbiology) within two working days, i.e. by Sep 09 2025 11:59PM.

Kind regards,

Christian

Christian Schnell, PhD

Senior Editor

PLOS Biology

cschnell@plos.org

---

## [Decision Letter · Decision Letter 1]

30 Oct 2025

Dear Dr Wang,

Thank you for your patience while we considered your revised manuscript "Neural computations of visual, semantic, and memorability features in the human brain" for publication as a Research Article at PLOS Biology. I am currently handling your manuscript while my colleague Christian Schnell is away from the office this week. Please accept my sincere apologies for the delays that you have experienced during this round of the peer review process. Your revised study has been evaluated by the PLOS Biology editors, the Academic Editor and two of the original reviewers.

In light of the reviews, which you will find at the end of this email, we would like to invite you to revise the work to thoroughly address the reviewers' reports.

As you can see, the reviewers agree that the revised version is strengthened and Reviewer #2 is now satisfied with the responses to the previous comments. However, Reviewer #1 continues to raise concerns about the analysis and interpretation of the data. Specifically, this includes the statistical models used to deal with multiple observations from the same subjects and that the HGP plots appear highly similar across brain regions.

Given the extent of revision needed, we cannot make a decision about publication until we have seen the revised manuscript and your response to the reviewers' comments. Your revised manuscript is likely to be sent for further evaluation by all or a subset of the reviewers.

**IMPORTANT - SUBMITTING YOUR REVISION**

*Re-submission Checklist*

*Published Peer Review*

*PLOS Data Policy*

*Blot and Gel Data Policy*

Best regards,

Richard

Richard Hodge, PhD

rhodge@plos.org

On behalf of:

Christian Schnell, PhD

cschnell@plos.org

REVIEWS:

Reviewer #1: In this revised manuscript, the authors have made substantial changes to the presentation, reporting and interpretation of their findings, including new data. These updates have strengthened the authors' claims, however, there remain key issues regarding anatomical and physiological treatment/interpretation of the data.

Major:

PFC results and nested data: As intracranial data is nested, i.e. contains multiple observations from the same individuals, it is standard practice to deal with this violation of observation independence via mixed effects designs or collapsing observations within individuals. This issue was raised for PFC findings because it was clear that only a subset of data/subjects (10 electrodes) were driving the statistical effect. The authors have responded to these concerns, but I'm not sure the key issue has been adequately addressed. Firstly, I appreciate the authors focused only on responsive electrodes and have added more PFC data. However, of the 59 "added" electrodes (6 subjects), what is important are how many were responsive and made it into the analysis, for which it appears 29 responsive probes were actually added (with 12 total being significant). I might not have followed correctly as the authors didn't make this clear, but only 2 additional/significant PFC electrodes were actually added. Secondly, the authors note that 7/20 subjects contributed significant PFC channels, suggesting the results are not driven by individual differences. However, this is actually the concern being raised, namely that less than half of the sample shows the reported effect, suggesting it's driven by a subset of subjects and doesn't strongly replicate as an observation. In relation to this, I appreciate that the authors have tried alternative statistical methods to suggest that controlling for subject nesting isn't necessary, but this is more a basic principle of statistical testing than an empirical question. A mixed-effects model, with subjects as a random factor, is required to appropriately deal with multiple observations from the same subjects. I do acknowledge that the authors have adjusted the strength of their claims regarding this result. Finally, the authors note some subjects had repeated recording sessions, it's not clear if this means during the same implantation or a second surgery. If this is just a second session during the same invasive monitoring, it should not be treated as new electrodes/observations, simply more trials for those individuals.

Response timing: It is expected that responses in visual regions occur before those in the MTL and PFC, following a wealth of data. The concern raised regarding the timing of responses was about the data itself, rather than methods of response time detection (onset/peak). As is clear from revision Figure 1, mean HGP plots appear highly similar across brain regions, with changes occurring at/before time zero. This suggests that HGP data has been heavily smoothed, and raises concerns for how such data across brain regions can look so similar. While the authors might highlight their focus on latency of the peak response, this is still being estimated from data that suggests brain-wide response onset across regions immediately to stimulus presentation, which makes little physiological sense.

Minor:

Model comparison: I appreciate the authors' expanding their modeling analysis. This is very helpful, particularly for the ResNet (visual) and SGPT (semantic) models. However, should we be surprised or read into the finding that multiple layers of SBERT and ResMem outperform or are equal to the final layer in terms of significant channel %? Is that because earlier layers rely more on visual features, so we should expect a greater percentage than later layers?

Reviewer #2: The authors have done a fantastic job at thoroughly addressing my comments and those of the other reviewers. I greatly appreciated the updated introduction that makes the theoretical contribution of the work clearer. The many additional analyses (e.g., using other models, running additional 7T analyses, etc) have also really solidified the strength of the findings. Overall this work will be an important and impactful contribution to the literature and our understanding of how vision and memory work and interact. Great work.

---

## [Decision Letter · Decision Letter 2]

2 Dec 2025

Dear Shuo,

Thank you for your patience while we considered your revised manuscript "Neural computations of visual, semantic, and memorability features in the human brain" for publication as a Research Article at PLOS Biology. This revised version of your manuscript has been evaluated by the PLOS Biology editors and the Academic Editor.

Based on our Academic Editor's assessment of your revision, we are likely to accept this manuscript for publication, provided you satisfactorily address the following data and other policy-related requests:

* We would like to suggest a slightly different title to improve readability and accessibility for our broad audience. Would any of these two titles work for you?

Characterization of the spatiotemporal representations of visual, semantic, and memorability features in the human brain

OR

Dissociable spatiotemporal representation of visual, semantic and memorability features in the human brain shape object recognition and memory formation

* Please add the links to the funding agencies in the Financial Disclosure statement in the manuscript details.

* Please include the approval/license number of the ethical approval for the experiments.

* Please include information in the Methods section whether the study has been conducted according to the principles expressed in the Declaration of Helsinki.

DATA POLICY:

Regardless of the method selected, please ensure that you provide the individual numerical values that underlie the summary data displayed in the following figure panels as they are essential for readers to assess your analysis and to reproduce it: 3D, 4BCDEGHIJLMNO, and 6ABCDFHI

* CODE POLICY

We expect to receive your revised manuscript within two weeks.

*Published Peer Review History*

*Press*

Sincerely,

Christian

Christian Schnell, PhD,

Senior Editor

cschnell@plos.org

PLOS Biology

Reviewer remarks:

Reviewer #1: I appreciate the author's additional responses to the prior round of review, I have no further comments.

---

## [Editor Report · Decision Letter 3]

19 Dec 2025

Dear Shuo,

Thank you for your patience while we considered your revised manuscript "Characterization of the spatiotemporal representations of visual, semantic, and memorability features in the human brain" for publication as a Research Article at PLOS Biology. This revised version of your manuscript including the updated source data file has been evaluated by the PLOS Biology editors.

Unfortunately, I still could not fully match the source data to the figure plots, and some of the plots were also different to the original plots in the paper. I have checked this only carefully for Figure 3D. Because our editorial office will be closed from Monday Dec 22nd to Jan 4th, I am returning the manuscript back to you for a careful check. Please have a look through the figures and make sure that the source data match the data presented in the figures. Please also provide a short rationale for any changes to the figures that need to be made, including the one you have already provided. I have not updated the figure yet, but you can do this now yourself.

We expect to receive your revised manuscript within three weeks.

*Published Peer Review History*

*Press*

Sincerely,

Christian

Christian Schnell, PhD

Senior Editor

cschnell@plos.org

PLOS Biology

---

## [Editor Report · Decision Letter 4]

7 Jan 2026

Dear Shuo,

Happy new year and thank you for the submission of your revised Research Article "Characterization of the spatiotemporal representations of visual, semantic, and memorability features in the human brain" for publication in PLOS Biology. On behalf of my colleagues and the Academic Editor, Christopher Pack, I am pleased to say that we can in principle accept your manuscript for publication, provided you address any remaining formatting and reporting issues. These will be detailed in an email you should receive within 2-3 business days from our colleagues in the journal operations team; no action is required from you until then. Please note that we will not be able to formally accept your manuscript and schedule it for publication until you have completed any requested changes.

PRESS

We frequently collaborate with press offices. If your institution or institutions have a press office, please notify them about your upcoming paper at this point, to enable them to help maximize its impact. If the press office is planning to promote your findings, we would be grateful if they could coordinate with biologypress@plos.org. If you have previously opted in to the early version process, we ask that you notify us immediately of any press plans so that we may opt out on your behalf.

Sincerely,

Christian

Christian Schnell, PhD

Senior Editor

PLOS Biology

cschnell@plos.org